# Paradoxical response reversal of top-down modulation in cortical circuits with three interneuron types

**Luis Carlos Garcia del Molino, Guangyu Robert Yang, Jorge F Mejias†, Xiao-Jing Wang\***

Center for Neural Science, New York University, New York, United States

**Abstract** Pyramidal cells and interneurons expressing parvalbumin (PV), somatostatin (SST), and vasoactive intestinal peptide (VIP) show cell-type-specific connectivity patterns leading to a canonical microcircuit across cortex. Experiments recording from this circuit often report counterintuitive and seemingly contradictory findings. For example, the response of SST cells in mouse V1 to top-down behavioral modulation can change its sign when the visual input changes, a phenomenon that we call response reversal. We developed a theoretical framework to explain these seemingly contradictory effects as emerging phenomena in circuits with two key features: interactions between multiple neural populations and a nonlinear neuronal input-output relationship. Furthermore, we built a cortical circuit model which reproduces counterintuitive dynamics observed in mouse V1. Our analytical calculations pinpoint connection properties critical to response reversal, and predict additional novel types of complex dynamics that could be tested in future experiments.

**\*For correspondence:** xjwang@nyu.edu

**Present address:** †Swammerdam Institute for Life Sciences, Center for Neuroscience, Faculty of Science, University of Amsterdam, Amsterdam, Netherlands

**Competing interests:** The authors declare that no competing interests exist.

## Introduction

Three major non-overlapping classes of interneurons expressing parvalbumin, somatostatin and vasoactive intestinal peptide (henceforth denoted PV, SST and VIP respectively) make up more than 80% of GABAergic cells of mouse cortex (*Rudy et al., 2011*). These neurons show cell-type-specific connectivity among themselves and with excitatory (E) neurons (*Pfeffer et al., 2013*; *Jiang et al., 2015*) forming a canonical microcircuit in the cortex. This microcircuit motif, initially proposed theoretically (*Wang et al., 2004*), has been the subject of numerous recent experimental studies using optogenetic tools applied to behaving mice (*Lee et al., 2012*; *Saleem et al., 2013*; *Kepecs and Fishell, 2014*; *Hawrylycz et al., 2016*) as well as computational studies (*Lee and Mihalas, 2015*; *Lee and Mihalas, 2017*; *Lee et al., 2017*; *Yang et al., 2016*; *Yang and Wang, 2017*). However, we still do not fully understand the mechanisms that underlie the behavior of this microcircuit which are often complex and counterintuitive.

A notable observation was that pyramidal neurons and VIP interneurons concomitantly increase their activities in the primary visual cortex V1 during locomotion in comparison with immobility (*Niell and Stryker, 2010*), even in the complete absence of visual input (*Keller et al., 2012*). Moreover, optogenetically activating (respectively inactivating) VIP interneurons mimics (respectively eliminates) the effect of running (*Fu et al., 2014*). Since VIP cells primarily target SST cells, a natural explanation for this phenomenon is disinhibition (*Wang et al., 2004*; *Lee et al., 2013*): activation of VIP cells suppresses SST cells, therefore neurons targeted by the SST population are disinhibited, enhancing the overall activity of excitatory neurons. However, recent experiments show that the network behavior might be more complex. Namely, in darkness the activation of VIP cells results in an average decrease of SST population activity (*Fu et al., 2014*), whereas in the presence of visual stimulation the response of SST cells is reversed and its firing rate increases during locomotion

compared to immobility (*Pakan et al., 2016*). These findings, which have been further confirmed in a recent preprint (*Dipoppa et al., 2017*), appear to challenge the disinhibition hypothesis, suggesting that the nature of the interaction between VIP and SST could be stimulus dependent.

These experimental results raise two questions: First, the external activation of a population that directly inhibits a second population can trigger a positive response of the latter. What is the mechanism behind this apparently paradoxical behavior? Second, the same top-down modulation can trigger both a positive response and a negative response of certain populations of the circuit depending on the sensory input. Under which conditions can we expect one response or the other?

In this study, we model cortical activity and provide a comprehensive answers to these two questions. We show that these counterintuitive phenomena rely on two basic features of cortical networks: (i) the presence of multiple populations of interneurons and (ii) nonlinear responses to input. Finally, we use our model to predict complex behaviors that have not yet been experimentally tested. Beyond the mechanistic explanation for the observed behavior in mice V1, our work provides a very general and powerful framework to explain the dynamics of neural networks with multiple interneuron types, their context-dependent interactions, and the emsergence of counterintuitive effects that may occur across different cortical structures and animals.

## Results

We simulate microcircuit activity using a four population firing rate model. The average rate of each population is given by a nonlinear function of its input that we refer to as the f-I curve (*Abbott and Chance, 2005*). The f-I curve is such that when the input is low (below threshold), cells are little responsive to changes in external input. Instead for high input (above threshold) small changes in the input can drive substantial changes in the response (*Miller and Troyer, 2002*) (see *Figure 1b*). This nonlinearity has been analyzed experimentally and theoretically (*Murphy and Miller, 2003*; *Phillips and Hasenstaub, 2016*) and as we will show later, it is a key feature of the model.

Populations are connected according to the microcircuit scheme in *Figure 1a* which contains the connections reported in both *Jiang et al., 2015* and *Pfeffer et al. (2013)*. We also consider three sources of input: (i) top-down modulation that targets VIP cells (ii) local recurrent input and (iii) constant background input set so that the populations have some fixed baseline activity (see Materials and methods for details).

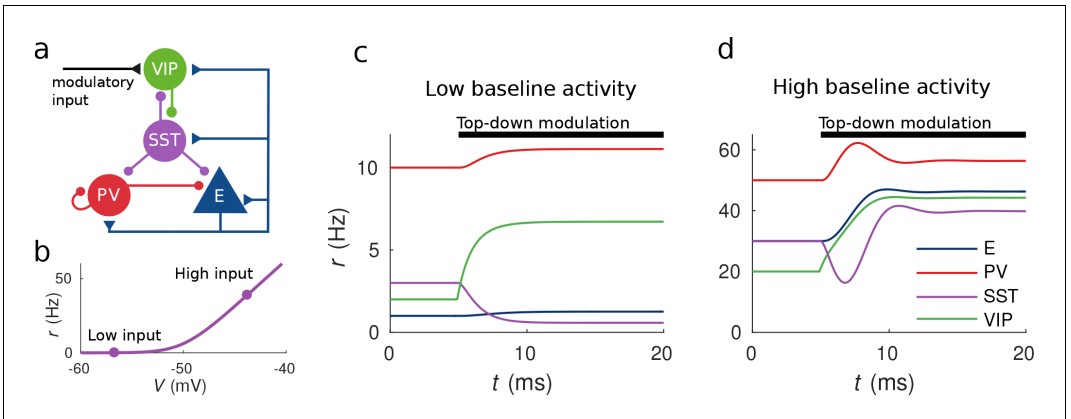

**Figure 1.** Response to top-down modulation depends on baseline activity. (**a**) Microcircuit connectivity and top-down modulatory input. (**b**) f-I curve. When input is low changes in input have almost no effect on the output rate, instead, when input is high changes in input have a big effect on output rate. (**c, d**) Transient dynamics upon the onset of the top-down modulatory current for low baseline activity (i.e. when the rates are low before top-down modulation) and high baseline activity (i.e. when the rates are high before top-down modulation). Under a low baseline activity condition, SST is inhibited and E and PV are slightly disinhibited. The high baseline activity condition shows an example of response reversal in SST activity: it initially goes below the baseline rate but due to significant change in E activity and to the recurrent excitation it eventually reverses to a rate higher than baseline.

## Response to top-down modulation depends on baseline activity

To illustrate possible complex behaviors displayed by the network, we first focused on the circuit responses to top-down modulation. The simulation results from our model allow us to identify two qualitatively different scenarios depending on the baseline activity of the network (the baseline activity is the activity before the onset of top-down modulation and we control it by changing the constant background input, see Materials and methods for details). On the one hand, when the baseline activity is low, top-down modulation will result in a decrease of the rate of the SST population and an increase of the rates of the other populations (E, PV and VIP) (see *Figure 1c*). On the other hand, when baseline activity is high, the rate of all populations increases with top-down modulation (see *Figure 1d*). These simulations reveal that population responses to top-down modulation depend in a complex way on the initial state of the network.

The striking behavior exhibited by the SST population can be explained heuristically by analyzing the response of the different populations to external excitatory input targeting VIP cells. When the top-down modulation starts, the rate of the VIP population increases. By calculating the time derivatives of the rates right after the onset of the top-down modulation (see Materials and methods) one can see that this effect always results in a transient reduction of SST activity and therefore a reduction of inhibition to VIP, PV and E cells. When baseline activity is low the E population is below threshold and this change in net input has a small effect in the output. In that situation, all populations quickly reach a stationary state. However, when the baseline activity is high, the E population is above threshold and a small change in input from SST cells has a big effect on the rate of the E population. If the recurrent excitation in the microcircuit is strong enough, it can reverse the initial response of the SST population making it increase its activity to a higher rate than the baseline.

## Circuit behavior explained by response matrix

In order to formally characterize the steady state response of a population to external input we introduce the response matrix $M$. The intuition behind the response matrix is that if we change the input to population $j$ (where $j = E, P, S, V$ for excitatory, PV, SST and VIP populations respectively) by a small amount $\delta I_j$, then the change in rate of the population $i$ will be $\delta r_i = \delta I_j M_{ij}$. If $M_{ij}$ is positive (negative), an increase of the external excitation to $j$ will result in an increase (decrease) of the rate of population $i$ (see Materials and methods and Table 3 for details). In contrast to the connectivity matrix, which takes into account only the direct path from population $j$ to $i$, the response matrix contains information about all the possible ways in which population $j$ can affect population $i$, namely through indirect connections $j$-$h$-$i$. Due to the complexity of these indirect pathways, for different values of the connectivity matrix (but preserving the excitatory/inhibitory structure) $M_{ij}$ can be positive or negative irrespective of whether the connection from $j$ to $i$ is inhibitory or excitatory. Furthermore, due to the nonlinearities in the f-I curve, the response depends on the baseline rate of each of the populations and, as shown before, it can reverse its sign.

As an example, we analyze in detail the response of the SST population to external input to VIP cells. As we show in the Materials and methods section, this term of the response matrix is given by:

$$M_{SV} = C w_{SV}((w_{EE} - d_E)(w_{PP} + d_P) - w_{EP}w_{PE}),$$

where $w_{ij}$ are the absolute values of the connection weights and therefore are positive by definition and for the system to be stable $C$ has to be positive (see Materials and methods for details). The terms $d_i$ are proportional to the inverse of the first derivative of the f-I curves and are always positive. In particular, $d_E$ becomes arbitrarily large when the input is very low and tends monotonically to a positive constant $d_E^\infty$ for high input. Therefore, if $w_{EE} \leq d_E^\infty$ then $M_{SV}$ will always be negative. However, for $w_{EE} > d_E^\infty$ the behavior is much richer: if input is high then $d_E$ will be close to its minimum $d_E^\infty$ and $w_{EE} > d_E$ allowing for $M_{SV}$ to be positive (provided that the product $w_{EP}w_{PE}$ is small enough). Instead if the input is low, $d_E$ will become very large and $M_{SV}$ will be negative.

It is remarkable that this change in the interaction between VIP and SST populations depends on the activation level of E: modifying the state of one population has a impact in the interactions between other populations. The heuristic explanation is that if the recurrent excitation is strong enough and the E population is already strongly excited (above threshold), a small decrease in the inhibition from SST to the E population can boost its activity and therefore strongly drive the whole

microcircuit. If instead, the E population is in a low activation state the change in inhibition will have a weak effect that will not be able to reverse the response of SST.

This observation provides an explanation to the reversal of the response of SST to VIP activation when the baseline activity is changed: as we show in *Figure 2a and c* for low baseline activity, $M_{SV}$ is negative and the presence of an external excitatory current targeting VIP cells will result in a negative response of SST cells and positive response of E, PV and VIP cells, conforming to the disinhibitory hypothesis. On the other hand, for high baseline activity (panels 2b and 2d), the response of the SST population to input to VIP cells becomes positive leading to the response reversal regime.

A similar analysis can be conducted for all terms in $M$. For example, another case of response reversal in this circuit is that of $M_{EE}$ which can have different signs for different baseline activity levels, meaning that the excitatory population can have a negative response to excitatory input to itself. Intuitively, if an external excitatory current targets the E population, its rate will increase transiently and thus the excitation that SST and VIP receive will also increase. If this effect is stronger in SST than in VIP the rate of the VIP population will decrease and therefore the inhibition that SST receives will decrease as well resulting in stronger inhibition to E cells. Note that for this to happen both SST and VIP have to be in the high activity baseline (i.e. $d_S$, $d_V$ have to be small) and $w_{SV}$, $w_{VS}$ have to be

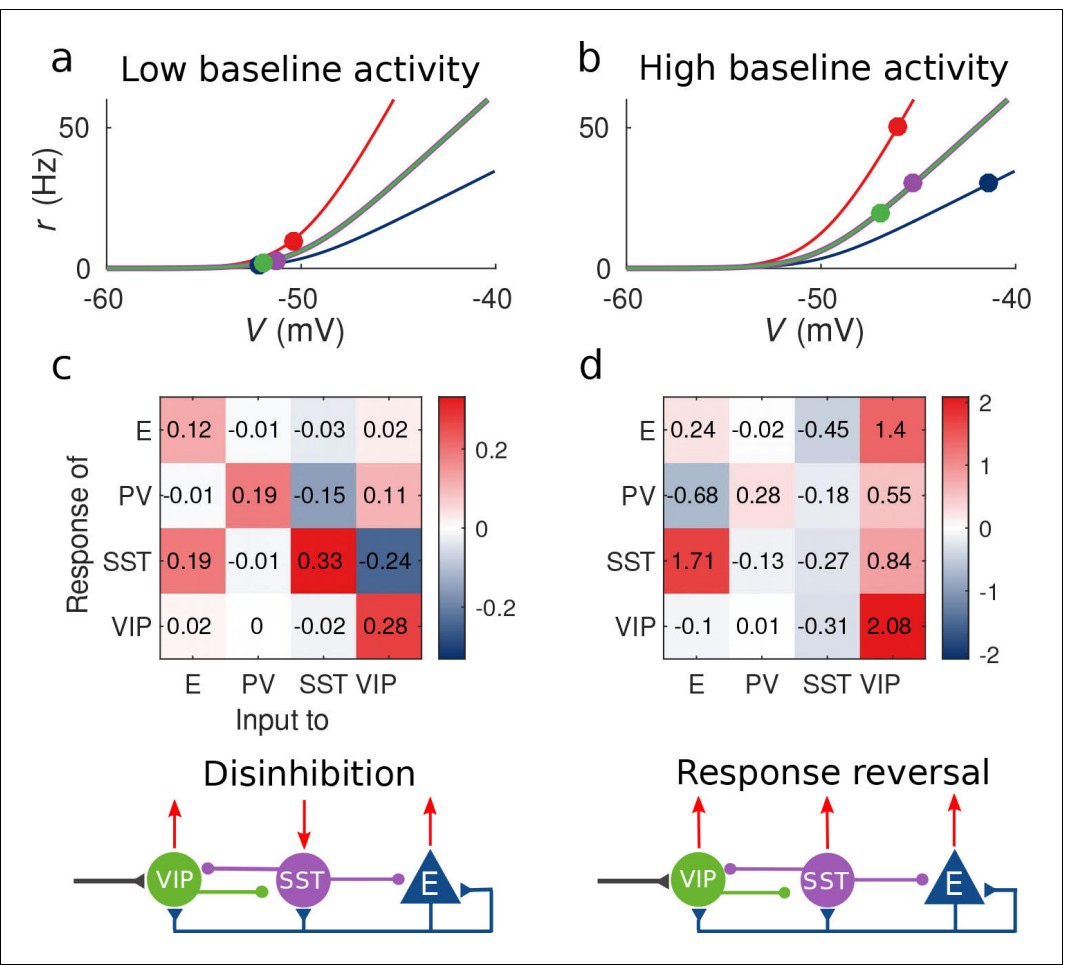

**Figure 2.** Response matrix and disinhibition vs. response reversal regime. (a–b) Tuning curves for the different populations and baseline activity in both scenarios (low and high). In the low baseline activity scenario (a) all populations are below threshold (flat part of the fI curve), instead in the high baseline activity scenario (b) all populations are above threshold, where small changes in input result in large changes in rate. (c–d) Response matrices for the two scenarios. In (c) the response of SST to external excitation of VIP is negative, while the responses of E and PV are positive. This corresponds to the disinhibition regime. In (d) the responses of all populations to external excitation of VIP are positive, in particular, the response of SST is reversed with respect to (c) corresponding to the response reversal regime.

strong. The explicit expression of $M_{EE}$ (see Table 3) reveals that if the SST-VIP-SST loop is not strong enough or if $d_S$, $d_V$ are large $M_{EE}$ will always be positive.

## Random network model

Experimental recordings showed a great diversity across neural responses even when recording from the same class of cells (Pyramidal, SST, PV or VIP) (*Pakan et al., 2016*). Although this diversity can have many origins, such as intrinsic heterogeneity in the cells within the same class, we proposed that random connectivity alone is sufficient to explain it. To do so we develop an extension of our model where each population is composed of multiple identical randomly connected rate units and where the probability that one connection exists from one unit to another depends on the populations of the presynaptic and postsynaptic units according to data extracted from *Jiang et al. (2015)*; *Pfeffer et al. (2013)* (see Materials and methods for details).

For each unit, we measure the rate modulation (rate during top-down modulation minus baseline activity) for the different baselines. If the rate modulation is positive it means that the neuron is more active in the presence of the modulatory current and vice versa. In *Figure 3*, we show scatter plots of the rate modulation under the low baseline condition versus the rate modulation under the high baseline condition for each unit. These simulations reveal that the behavior of individual neurons can be quite variable while the population average still corresponds to the behavior of the population-based model. Since all units of each population are identical, variability in the response has to be due to the heterogeneity in the connectivity. This variability can result in cells within the same population having responses with opposite sign, as has been observed to be the case in mouse V1 (*Reimer et al., 2014*; *Pakan et al., 2016*) and A1 (*Kuchibhotla et al., 2017*). In addition, variability might also have further implications for gating of signals, since variability in inhibitory cells has been proposed to modulate the response gain of neural circuits (*Mejias and Longtin, 2014*).

## Model of mouse V1 accounts for experimental measurements

Our framework allows us to easily understand the counterintuitive behavior of V1 during locomotion. In the experiments mice with their head fixed face a screen where different visual stimuli are presented and can run freely on a treadmill (*Fu et al., 2014*; *Pakan et al., 2016*). Different visual stimuli result in different baseline activities in V1 and top-down modulation is triggered when the mice start running.

To model visual input we use external currents. In the case of size-varying gratings, this input has two sources: thalamic input that targets excitatory cells and cortical input that targets SST cells. In order to reproduce the surround suppression effect (*Ozeki et al., 2009*; *Adesnik et al., 2012*), excitatory cells have a small receptive field and therefore receive center input and SST cells have a large receptive field and receive surround input (see Materials and methods for details).

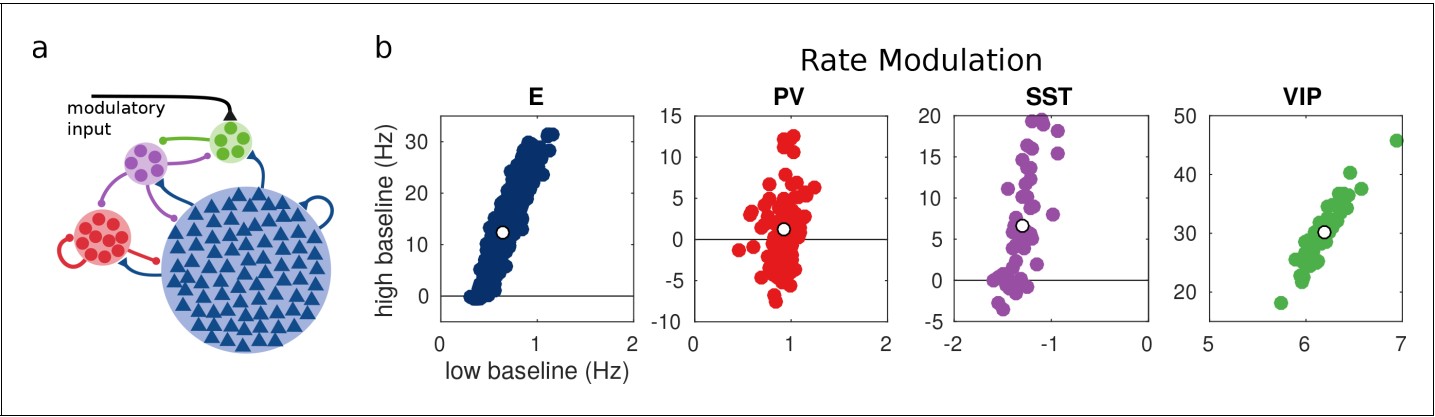

**Figure 3.** Random network model. (a) Schematic of the model. Each population is composed of several rate units and the connectivity between units is random with probabilities extracted from experimental data in the literature. (b) Rate modulation (rate after the onset of the modulatory current minus baseline rate) for low and high baseline activities. Each colored point corresponds to one unit. Unit responses are very variable and, in particular within the same population different units might have responses with different sign. White points correspond to the population average. Despite the variability of individual responses the population average corresponds to the population responses in the single unit model in *Figure 1*.

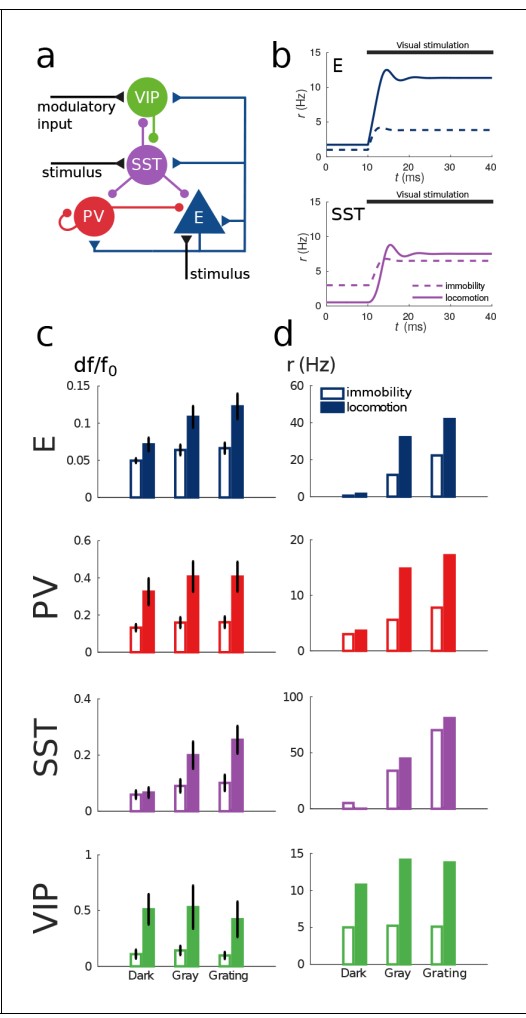

**Figure 4.** Model of mouse V1 behavior. (**a**) Schematic of the microcircuit. Visual input targets E and SST cells. Behavior related top-down modulation targets VIP cells. (**b**) Response of E and SST populations when a weak visual stimulus (6 deg) is presented for locomotion and immobility. The E population always shows a higher response with locomotion. On the other hand, before the visual stimulation the SST population has higher activity for immobility than for locomotion and when the visual stimulus is presented, the activity of the SST population is higher for locomotion. (**c**) Relative change in calcium fluorescence for three levels of visual stimulation (darkness, gray screen and grating) and two behavioral states: immobility (empty bars) and locomotion (filled bars) extracted from *Pakan et al. (2016)*. (**d**) Rates (in Hz) of the populations in the V1 simulation for the same conditions as in (**c**). Comparison of (**c**) with (**d**) shows that our simulations reproduce qualitatively the activity of neural populations in mice V1. Namely the activity of all populations is higher during locomotion than during immobility whenever there is visual stimulation and for E, PV and VIP also in the absence of visual stimulation. Our model shows a decrease in activity of SST during locomotion as reported in *Fu et al. (2014)* (the change in activity of the SST population in darkness in *Pakan et al. (2016)* is not statistically significant). The quantitative differences might be related to the fact that changes in calcium fluorescence are not proportional to changes in rate.

The online version of this article includes the following figure supplement(s) for figure 4:

**Figure supplement 1.** Model of mouse V1 behavior with different grating sizes.
**Figure supplement 2.** Robustness of the behavior.
**Figure supplement 3.** Alternative architectures.

*Figure 4b* shows the response reversal phenomenon when a weak visual stimulus is presented. Before the visual stimulation, the SST has higher activity for immobility than for locomotion, by contrast, when the visual stimulus is presented, the activity of the SST population is higher for locomotion. In *Figure 4c*, we show the experimental data from *Pakan et al. (2016)* for three different experimental conditions (darkness, gray screen and grating) and in *Figure 4d* our simulations of V1 under the same conditions. *Figure 4—figure supplement 1* shows the experimental data from the preprint (*Dipoppa et al., 2017*) for gratings of different sizes alongside with the behavior of our model.

Our simulations of this V1 circuit model reproduce the phenomena described in the literature: in the presence of visual stimulation, the activities of all populations, including SST, increase during locomotion (*Pakan et al., 2016*). In darkness, the activities of excitatory, PV and VIP populations increase during locomotion while the activity of SST decreases as reported in *Fu et al. (2014)* and in the preprint (*Dipoppa et al., 2017*). In *Pakan et al. (2016)*, the response of SST to locomotion in darkness is weakly positive but this result is not statistically significant while the other two are.

To show that our results do not rely on a fine tuning of the connectivity parameters or even on certain details of the microcircuit structure, we have run the model with several connectivity matrices and perturbations of them (*Figure 4—figure supplement 2*) and we find that different connectivity parameters can reproduce the same circuit behavior as has been shown before in other systems (*Marder et al., 2015*). We have also considered other microcircuit structures to account for the differences between studies ([*Pfeffer et al., 2013*] reports projections from PV to VIP and (*Jiang et al., 2015*) from PV to SST) and we also consider thalamic input to PV (*Figure 4—figure supplement 3*). In all these cases, the results were consistent with our original findings showing that the phenomenon and the analysis are robust and not a peculiarity of one specific circuit.

## Discussion

We have developed a theoretical model of cortical circuit with multiple interneuron types that accounts for newly identified complex interactions between cell types. The model has been used to reproduce and explain two counterintuitive phenomena observed in mouse cortex. First, in certain cases the activation of VIP cells results in an overall positive response of the SST population (*Pakan et al., 2016*). Second, the sign of the SST population response to excitation of VIP cells depends on the baseline activity of the circuit (*Fu et al., 2014*). Two features of the system lead to this behavior: the presence of multiple interneuron populations and the nonlinearity of f-I curves.

We explained heuristically the response reversal by closely looking at transient dynamics of the circuit. One experimentally testable prediction of our analysis is that, as *Figure 1d* and our calculations of the transient behavior show, in the response reversal regime, the overall SST population response to top-down modulation should initially decrease and later increase until reaching a higher rate than the baseline.

Based on our model, we introduced the response matrix $M$, which is a comprehensive framework to understand counterintuitive steady state responses. It provides explicit information about the contribution of each individual connection. For example by looking at the elements in $M_{SV}$ (see Table 3), one can readily see that if the recurrent excitation between pyramidal cells is not large enough, $M_{SV}$ can only be negative and therefore response reversal of SST would not happen. This statement can be easily tested by repeating the experiments while suppressing the activation of the E population. As we discussed before, another example is that if both SST and VIP populations have high baseline activities and if the SST-VIP-SST loop is strong enough, $M_{EE}$ can be negative, that is the excitatory population can have a negative response to excitatory input (see Table 3 for the explicit expression of $M_{EE}$). If the connections between the SST and the VIP populations are removed (or weakened) or if their baseline activities are sufficiently lowered $M_{EE}$ will always be positive. This constitutes another interesting prediction that can be experimentally tested.

Our calculations also revealed sign correlations between entries of $M$, for example $M_{SV}$ and $M_{SS}$ have opposite signs for any connectivity matrix (given the microcircuit) and for any baseline activity. This predicts that in the regime where SST activity has a positive response to excitatory input targeting VIP, SST has to have a negative response to external input targeting SST. In addition, our results are in line with experimental studies that show that VIP interneurons play an important role in cortical activity modulation (*Mesik et al., 2015*; *Ibrahim et al., 2016*; *Jackson et al., 2016*).

Our approach constitutes a general conceptual framework in which previous work regarding complex cortical interactions can be better understood (*Tsodyks et al., 1997*; *Ozeki et al., 2009*; *Litwin-Kumar et al., 2016*). The analysis of the response matrix shows that for the given microcircuit structure all terms of the matrix can be positive or negative. This is not the case in E-I networks (networks with one excitatory (E) population and only one inhibitory (I) population) (*Tsodyks et al., 1997*; *Ozeki et al., 2009*). In that case $M_{EE}$ and $M_{IE}$ are always positive, $M_{EI}$ is always negative and only $M_{II}$ can have both signs (see Materials and methods). In this sense, having more than one inhibitory population results in a much more versatile network. Another important point that can be

derived from our calculations is the relationship between response reversal and inhibition stabilized networks (ISN) (*Ozeki et al., 2009*). Looking at the terms of the response matrix for an E-I network, we can see that the condition to have response reversal and the condition to be an ISN is the same: $W_{EE}$ has to be larger than $d_E^\infty$. When analysing networks with more than one inhibitory population the relationship is not necessarily bidirectional any more. In the network that we analyzed, we found that in the high baseline activity the network is in the ISN regime and $M_{SV}$ is positive (as observed in [*Litwin-Kumar et al., 2016*]), whereas in the low baseline activity the network is not in the ISN regime and $M_{SV}$ is negative, so in this case there is a clear relationship between being an ISN and exhibiting response reversal. However, the condition for other cases of response reversal such as $M_{EE}$ do not involve $W_{EE}$ and therefore do not require the network to be an ISN.

Finally, this study provides a parsimonious yet powerful explanation to striking observations of interneuronal circuits in V1 (*Fu et al., 2014*; *Pakan et al., 2016*; *Lee et al., 2017*) without requiring the assumption of top-down excitatory inputs explicitly targeting SST or PV neurons. Both our computational neural network model and the approach presented here (the response matrix analysis) go beyond circuit dynamics in mice V1 and can be easily applied to other species and cortical areas. By extending previous works (*Tsodyks et al., 1997*; *Ozeki et al., 2009*), it naturally explains the response reversal observed in cat visual cortex (*Ozeki et al., 2009*). It could also be applied to explain similar phenomena observed in mouse primary auditory cortex (*Seybold et al., 2015*; *Kuchibhotla et al., 2017*). In particular, in *Kuchibhotla et al. (2017)*, the authors find that locomotion reduces the activity of excitatory cells. Assuming that the main modulation in the circuit is mediated by VIP cells this observation implies that $M_{EV}<0$ which is the case when the connections $W_{EP}$ and $W_PS$ are strong enough. In mouse somatosensory cortex, activating VIP neurons results in an intuitive decrease in SST activity, instead of a response reversal (*Lee et al., 2013*). As our results suggest, this qualitative difference between V1 and somatosensory cortex may be explained by the quantitative difference between their circuit architectures: in a recent study the authors showed that cell densities of different types of interneurons differ substantially across cortical areas resulting in counterintuitive impacts on circuit responses (*Kim et al., 2017*). These responses can be readily understood using the response matrix.

In this work, we mainly focused on steady-state responses. However, neural responses in many cortical areas, including primary auditory cortex, are largely transient and dynamical (*Wehr and Zador, 2003*). In addition, synaptic connections to and from interneurons are often subject to short-term plasticity (*Reyes et al., 1998*). Understanding transient dynamics in nonlinear, multi-type interneuronal circuits would be an important topic for future research.

We have shown that similar to the now well-known paradoxical effect that the presence of a single inhibitory neuron type can cause (*Tsodyks et al., 1997*; *Ozeki et al., 2009*), the presence of multiple types of interneurons has an even stronger impact on the activity of neural circuits. We have also exposed the effect of nonlinearity of the f-I curve. Our analysis suggests that in a circuit with multiple populations, the most interesting circuit behavior is found when spontaneous baseline activity is close to threshold since in that regime responses will change the most with small changes in population rates. These two features significantly broaden the richness of the dynamics of cortical circuits and enhance their usefulness for cognitive and behavioral computations. We conclude that computational models and mathematical analysis are critical to fully understand the dynamics of neural circuits underlying behavior (*Gjorgjieva et al., 2016*), especially when several types of interneurons are involved as intuition alone may be misleading and provide erroneous predictions on such circuits.

## Materials and methods

### Firing-rate-based population model

The state of the system is characterized by the rates $r_i$. To model the average rate of each population we use a function of the input $V_i$ as the one introduced in *Abbott and Chance (2005)*

$$r_i = f(V_i) = \frac{V_i - V_{th}}{\tau(V_{th} - V_r)}\frac{1}{1 - e^{-(V_i - V_{th})/v}} \qquad (1)$$

where $V_{th} = -50$ mV and $V_r = -60$ mV are the threshold and reset potentials respectively, $\tau$ is the

**Table 1.** Connectivity matrix (in pAs).

| | | From | | | |
|---|---|---|---|---|---|
| | | E | PV | SST | VIP |
| to | E | 2.42 | −0.33 | −0.80 | 0 |
| | PV | 2.97 | −3.45 | −2.13 | 0 |
| | SST | 4.64 | 0 | 0 | −2.79 |
| | VIP | 0.71 | 0 | −0.16 | 0 |

membrane time constant and $v = 1$ mV. $V_i$ is the average input to each of the populations and is given by

$$V_i = V_l + \left( \sum_j W_{ij} r_j + I_i + I_{bkg}^i \right) / g_l^i \tag{2}$$

where $V_l = -70$ mV is the reversal potential and $g_l^i$ is the membrane conductance. $W$ is the connectivity matrix and therefore $\sum_j W_{ij} r_j$ is the recurrent local input. $I_i$ is the external input current and $I_{bkg}^i$ is a constant current that is tuned to obtain the desired baseline activity and we find the specific values by solving the system $r_i = f(V_l + (\sum_j W_{ij} r_j + I_i + I_{bkg}^i) / g_l^i)$. For example, for the baseline activity steady-state the background currents needed to obtain the desired rates (1, 10, 3 and 2 Hz for pyramidal, PV, SST and VIP, respectively) are 114.7, 233.6, 94.3 and 89.9 pA. The rate dynamics are given by

$$\tau_r \frac{dr_i}{dt} = -r_i + f(V_i) \tag{3}$$

where $\tau_r = 2$ ms (**Gerstner, 2000**). Since the parameters of the f-I curve are population dependent (see Table 2), different populations will have different rates for the same input. The nonlinearity of the f-I curve has very important consequences. Namely, for low input $f(V_i)$ is almost flat, and therefore changes in the input will have almost no effect on the rate. By contrast, for strong input $f(V_i)$ tends asymptotically to a straight line with slope $\frac{1}{\tau_i(V_{th} - V_r)}$ and changes in the input will elicit a large change in the rate. As we will show later, this feature is key to reproduce the response reversal observed in the experiments.

The connectivity matrix $W$ used in the simulations is generated by rejection sampling, that is by generating random matrices that have the microcircuit structure given in **Figure 1a** and selecting the ones that produce the desired responses. The simulations of **Figures 1** and **2** were done with the connectivity matrix given in **Table 1**.

Behavioral state is modeled with a constant top-down modulatory current of 10 pA that targets VIP cells. The constant background inputs $I_{bkg}^i$ are set so that in the absence of the top-down modulatory current, the E, PV, SST and VIP populations will have spontaneous average rates of 1, 10, 3 and 2 Hz, respectively, for the low baseline activity scenario and 30, 50, 30 and 20 Hz for the high baseline activity.

## Time derivatives of the rates after the onset of modulation

In this section, we calculate analytically the changes in rate right after the onset of the modulatory current. The intuition behind these calculations is that the initial change in activity of a population is driven by the fastest path from the external input to the neurons in that population.

We assume that the system is at a fixed point (therefore $\frac{dr_i}{dt} = 0$ for all populations) and that at time $t = 0$ an excitatory top-down modulatory current targets the VIP population. Taking into

**Table 2.** Population-dependent parameters.

| | E | PV | SST | VIP |
|---|---|---|---|---|
| $g_l$ | 6.25 nS | 10 nS | five nS | five nS |
| $\tau$ | 28 ms | 8 ms | 16 ms | 16 ms |

**Table 3.** Entries of the respone matrix.

| |
|---|
| $M_{EE} = C(w_{PP} + d_P)(d_S d_V - w_{SV} w_{VS})$ |
| $M_{PE} = C(w_{PE}(d_S d_V - w_{SV} w_{VS}) - w_{PS}(w_{SE} d_V - w_{SV} w_{VE}))$ |
| $M_{SE} = C(w_{PP} + d_P)(w_{SE} d_V - w_{SV} w_{VE})$ |
| $M_{VE} = C(w_{PP} + d_P)(w_{VE} d_S - w_{SE} w_{VS})$ |
| $M_{EP} = -C w_{EP}(d_S d_V - w_{SV} w_{VS})$ |
| $M_{PP} = -C((w_{EE} - d_E)(d_S d_V - w_{SV} w_{VS}) + w_{ES}(w_{SE} d_V - w_{SV} w_{VE}))$ |
| $M_{SP} = -C w_{EP}(w_{SE} d_V - w_{SV} w_{VE})$ |
| $M_{VP} = -C w_{EP}(w_{VE} d_S - w_{SE} w_{VS})$ |
| $M_{ES} = -C d_V(w_{ES}(w_{PP} + d_P) - w_{EP} w_{PS})$ |
| $M_{PS} = -C d_V(w_{ES} w_{PE} - (w_{EE} - d_E) w_{PS})$ |
| $M_{SS} = -C d_V((w_{EE} - d_E)(w_{PP} + d_P) - w_{EP} w_{PE})$ |
| $M_{VS} = -C(w_{VE}(w_{ES}(w_{PP} + d_P) - w_{EP} w_{PS}) + w_{VS}((w_{EE} - d_E)(w_{PP} + d_P) - w_{EP} w_{PE}))$ |
| $M_{EV} = C w_{SV}(w_{ES}(w_{PP} + d_P) - w_{EP} w_{PS})$ |
| $M_{PV} = C w_{SV}(w_{ES} w_{PE} - (w_{EE} - d_E) w_{PS})$ |
| $M_{SV} = C w_{SV}((w_{EE} - d_E)(w_{PP} + d_P) - w_{EP} w_{PE})$ |
| $M_{VV} = C(w_{ES}(w_{ES}(w_{PP} + d_P) - w_{EP} w_{PS}) - d_S((w_{EE} - d_E)(w_{PP} + d_P) - w_{EP} w_{PE}))$ |

account that the time derivatives of the rates are given by *Equation (3)* and since $f(V)$ is monotonously increasing and the modulatory current $I_V > 0$, then $\frac{dr_V}{dt}(0)$ will be positive and all other derivatives will still be 0. In order to estimate the behavior of the initial slope of $\frac{dr_i}{dt}$, we calculate the second derivatives at $t = 0$:

$$
\begin{aligned}
\frac{d^2 r_i}{dt^2} &= \frac{1}{\tau_i}\frac{d}{dt}\left(-r_i + f(V_i)\right) \\
&= \frac{1}{\tau_i}\left(-\frac{dr_i}{dt} + \frac{df}{dV_i}\sum_j \frac{dV_i}{dr_j}\frac{dr_j}{dt}\right) \\
&= \frac{1}{\tau_i}\left(-\frac{dr_i}{dt} + \frac{df}{dV_i}\frac{W_{iV}}{g_l^i}\frac{dr_V}{dt}\right)
\end{aligned}
\tag{4}
$$

where in the last step we used the fact that $\frac{dr_i(0)}{dt} = 0$ except for VIP. Since $\frac{df}{dV_i}$, $g_l^i$ and $\frac{dr_V}{dt}$ are positive, the sign of $\frac{d^2 r_i}{dt^2}$ will depend on the sign of $W_{iV}$. In particular, for SST we obtain

$$
\frac{d^2 r_S}{dt^2} = \frac{1}{\tau_S}\frac{df}{dV_S}\frac{W_{SV}}{g_l^S}\frac{dr_V}{dt}(0) < 0,
\tag{5}
$$

meaning that in all regimes the initial (transient) response of the SST population to top-down modulation targeting VIP cells will be negative.

## Response matrix and response reversal

In order to characterize the response of a population to external excitatory input to the network we calculate how its rate will change for a small change in external input. We focus on stationary states $r_i = f(V_i)$. If we apply a small perturbation to the external input $\delta I_i$, the network will reach a new stationary state

$$
r_i + \delta r_i = f(V_i + \delta V_i) = f(V_i) + f'(V_i)\delta V_i + O(\delta V_i^2)
\tag{6}
$$

where $f'(V_i)$ is the derivative of $f$ with respect to $V$ and

$$
\delta V_i = \left(\sum_j W_{ij}\delta r_j + \delta I_i\right)/g_l^i.
\tag{7}
$$

Since $r_i = f(V_i)$, when we linearize $f$ around $V$ and ignore terms of order $\delta V^2$ and higher we obtain the following self-consistent equation

**Table 4.** Connection probabilities for the random network model.

|    |     | From |    |     |     |
|----|-----|------|----|-----|-----|
|    |     | E | PV | SST | VIP |
|    | E   | 0.02 | 1 | 1 | 0 |
|    | PV  | 0.01 | 1 | 0.85 | 0 |
|    | SST | 0.01 | 0 | 0 | −0.55 |
| to | VIP | 0.01 | 0 | 0.5 | 0 |

$$\delta r_i = f'(V_i)\left(\sum_j W_{ij}\delta r_j + \delta I_i\right)/g_l^i. \tag{8}$$

We define the entries of response matrix as the derivative $M_{ij} = \frac{\partial r_i}{\partial I_j}$, which can be obtained from the limit $\delta I_j \to 0$ in the system of equations given by (*Equation 8*) and in matrix form can be written as

$$M = (D - W)^{-1} \tag{9}$$

where $D$ is a diagonal matrix with entries $D_{ii} = g_{l,i}/f'(V_i)$. As it was explained in the results section, the nonlinear behavior of the terms $D_{ii}$ is essential to explain the response reversal regime. $D_{ii}$ becomes arbitrarily large as $V_i \to -\infty$ and decreases monotonically to $d_i^\infty = \tau_i(V_{th} - V_r)/g_l^i$ when $V_i \to \infty$.

In Table 3, we give the explicit formulas to all the entries of the response matrix in terms of the entries of the connectivity matrix $W$ and $D$ (we denote $w = |W|$, $d_i = D_{ii}$ and $C = det(D - W)^{-1}$). Note that, because of the complex interactions in the network, the sign of $M_{ij}$ is never determined exclusively by that of $W_{ij}$.

## Random network model

We consider a network with 800 E units, 100 PV units, 50 SST units and 50 VIP units. Each unit within a population has the same f-I curve with the parameters in *Table 2*. The probabilities $p_{ij}$ of a connection from each unit in population $j$ to each unit in population $i$ are estimated from data (*Pfeffer et al., 2013*; *Jiang et al., 2015*) and are given in *Table 4*.

The strengths of the connections are rescaled so that the average input of a unit in population $i$ from all units in population $j$ is $W_{ij}$ as given in *Table 1*. More specifically, each unit in population $i$ will receive in average $m_{ij} = p_{ij}N_j$ projections from population $j$ (where $N_j$ is the number of units in population $j$) and therefore the weight of these connections will be $W_{ij}/m_{ij}$.

Top-down modulatory current and background input is identical to all units within the same population and has the same value as in the population based model.

## Mouse V1 model

In the simulations of V1 activity, we use the connectivity matrix given in *Table 5*.

We model visual input with an external excitatory current that targets E and SST cells. In the experiments in *Pakan et al. (2016)* and in the preprint (*Dipoppa et al., 2017*) the authors consider

**Table 5.** Connectivity matrix for the mouse V1 model (in pAs).

|    |     | From |    |     |     |
|----|-----|------|----|-----|-----|
|    |     | E | PV | SST | VIP |
| to | E   | 3.30 | −3.48 | −2.98 | 0 |
|    | PV  | 1.73 | −4.25 | −1.07 | 0 |
|    | SST | 3.50 | 0 | 0 | −4.51 |
|    | VIP | 0.53 | 0 | −0.13 | 0 |

three levels of visual stimulation which are: darkness, gray screen and grating. To model darkness condition, we assume a total absence of visual stimulation (therefore $I_E = 0$ pA, $I_S = 0$ pA). For gray screen, we use a small input current to the excitatory population ($I_E = 50$ pA, $I_S = 0$ pA). Finally to model different grating diameters the value of the input is a sigmoid function of the grating diameter $\theta$:

$$I_i(\theta) = \frac{a_i}{1 + e^{-\theta/b_i + 5}} \tag{10}$$

where $b_E = 2$, $b_S = 6$, $a_E = 100$ pA, $a_S = 20$ pA. With this parameters E cells receive center input (input saturates for diameters $\sim$20 deg) and SST cells receive surround input (input to SST saturates for diameters of $\sim$60 deg) (*Dipoppa et al., 2017*).

To demonstrate that our results do hold for a wide range of connectivity matrices and do not have to be fine tuned, we simulate several different connectivity matrices that produce the same qualitative behavior. We also make perturbations of these matrices by multiplying each entry by a random variable uniformly distributed in the interval $[0.9, 1.1]$. This amounts to randomly modifying each connection within $\pm$10% of its original value (see *Figure 4—figure supplement 2*).

In the alternative models of *Figure 4—figure supplement 3* where visual stimulus input also targets PV cells, we use $I_P = 0$ pA for darkness, $I_P = 10$ pA for gray screen and $b_P = 2$, $a_P = 20$ pA for gratings.

### Response matrix for an E-I network

For the sake of completeness, here we analyze the response matrix for a fully connected E-I network (*Tsodyks et al., 1997*, *Ozeki et al., 2009*) . The connectivity matrix is

$$W = \begin{bmatrix} w_{EE} & -w_{EI} \\ w_{IE} & -w_{II} \end{bmatrix} \tag{11}$$

and therefore the response matrix is

$$M = (D - W)^{-1} = C \begin{bmatrix} w_{II} + d_I & -w_{EI} \\ w_{IE} & -w_{EE} + d_E \end{bmatrix}, \tag{12}$$

where $C = ((d_E - w_{EE})(w_{II} + d_I) + w_{EI}w_{IE})^{-1}$. Note that the only term that can change sign is $M_{II}$ so the only population that can exhibit response reversal is the $I$ population. Furthermore, note that the condition for having response reversal ($w_{EE} > d_E^\infty$) is the same that defines the ISN regime, so this two properties are equivalent in the E-I network.

## Acknowledgements

This work was supported by the NIH grant R01MH062349, the ONR grant N00014-17-1-2041, STCSM grants 14JC1404900 and 15JC1400104.

## Additional information

### Funding

| Funder | Grant reference number | Author |
| --- | --- | --- |
| Office of Naval Research | N00014-17-1-2041 | Xiao-Jing Wang |
| Science and Technology Commission of Shanghai Municipality | 14JC1404900 | Xiao-Jing Wang |
| NIH Blueprint for Neuroscience Research | R01MH062349 | Xiao-Jing Wang |
| Science and Technology Commission of Shanghai Municipality | 15JC1400104 | Xiao-Jing Wang |

The funders had no role in study design, data collection and interpretation, or the decision to submit the work for publication.

## Author contributions

Luis Carlos Garcia del Molino, Conceptualization, Formal analysis, Investigation, Writing—original draft, Writing—review and editing; Guangyu Robert Yang, Jorge F Mejias, Conceptualization, Writing—original draft, Writing—review and editing; Xiao-Jing Wang, Conceptualization, Funding acquisition, Writing—review and editing

## Author ORCIDs

Luis Carlos Garcia del Molino http://orcid.org/0000-0001-9934-9461
Guangyu Robert Yang http://orcid.org/0000-0002-8919-4248
Jorge F Mejias http://orcid.org/0000-0002-8096-4891
Xiao-Jing Wang http://orcid.org/0000-0003-3124-8474

## Decision letter and Author response

Decision letter https://doi.org/10.7554/eLife.29742.sa1
Author response https://doi.org/10.7554/eLife.29742.sa2

# Additional files

## Supplementary files

• Transparent reporting form

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
