## [Decision Letter]

Thank you for submitting your article "Paradoxical response reversal of top-down modulation in cortical circuits with three interneuron types" for consideration by *eLife*. Your article has been favorably evaluated by Timothy Behrens (Senior Editor) and three reviewers, one of whom is a member of our Board of Reviewing Editors. The reviewers have opted to remain anonymous.

The reviewers have discussed the reviews with one another and the Reviewing Editor has drafted this decision to help you prepare a revised submission.

Summary:

This manuscript presents some important insights into the diverse, counterintuitive behaviors of circuits with interacting inhibitory neuron populations. The authors show that, in a circuit with three types of interneuron, the functional sign of interactions can change depending on the exact activity level of the different cell types in the network – a population that inhibits another in one regime may suppress it in another regime. The essential features to enable this are: 1) more than one type of interneuron, 2) with diverse thresholds/nonlinearities. They relate this result to the experimental literature through citations, and directly compare their model results to data figures from other authors. In the discussion, they give testable predictions.

Essential revisions:

1) In the standard mode (van Vreeswijk and Sompolinsky, Neural Computation 1998), connectivity is high, and so the diagonal terms, d_i_, are large. In this regime, there is no response reversal. It's an open question exactly how high connectivity is; it certainly isn't infinity, which is what physicists would like it to be, and the effective strength of the connectivity drops as the firing rate drops. However, when firing rates drop, fluctuations become important, and firing rate models become less believable. We're not asking the authors to do full network simulations (although we would suggest that it would be an interesting avenue for future research). However, they should at least comment on this. Even better would be a back of the envelope calculation showing that the connection strengths between populations are in the right range.

2) The authors do a good job citing the relevant literature. However they avoid framing their work in the context of inhibitory stabilized networks (ISNs). ISNs have very strong recurrent excitation that needs to be stabilized by recurrent inhibition (Ozeki et al.), and show the signature of a complex transient before settling in the equilibrium state – reminiscent of the author's Figure 1D. As remarked in the manuscript the sign flip of M_SV_ requires w_EE_ to be sufficiently large. Is the network an ISN? Figure 2 of Litwin-Kumar et al. (2016) extends ISNs to circuits with multiple interneurons subtypes, and they show that if the total inhibition received by E cells reduces under VIP stimulation then the network is an ISN. What regime is the author's model in? Is this a useful label for their network?

3) In Figure 2D, M_EE_ is negative (-0.35), and if we understand things correctly, it's always negative in the high gain regime. Thus, in the high baseline activity state, an input to the pyramidal neuron population will result in a decrease of pyramidal neuron firing rates. This is at odds with most (all?) data sets. The authors remark on this feature in the last paragraph of the subsection “Circuit behavior explained by response matrix”, but do not address the plausibility of this prediction. Do the authors think this result is a problem for their model? More generally, with new parameters can the authors explain the sign flip in M_SV_ without a sign flip in M_EE_ or are these tethered together somehow?

4) Dipoppa et al. 2016 is important for justifying the model and the authors cite it frequently (and even republish some of its figures). But this paper has not been peer-reviewed (it's a Biorxiv report), giving it the same veridical status as a personal communication or SFN abstract. It's not appropriate for a peer-reviewed manuscript to depend on data that has not been reviewed. In addition, we're not sure how this will affect Dipoppa et al.'s attempts to get their work published. *eLife* is peer reviewed, and many journals won't let you republish work that's already published in a peer reviewed journal. In this manuscript, the authors actually take figures out of the other group's non-reviewed preprint and publish them in their own paper.

It seems to us that Dipoppa et al. is not absolutely essential; Figure 4E could be dropped without affecting the paper much. If the authors do want to include it, they should do two things. First, they should make it crystal clear that Dipoppa et al. is not peer-reviewed, every single time the citation is made. They can leave no doubt in the readers' minds that data is not yet part of the scientific literature. Second, they should get permission from Dipoppa et al. before publishing their data. We're guessing *eLife* requires this, but even if it doesn't, it's not worth irritating one's colleagues for something that is not essential to one's story.

---

## [Author Response]

Essential revisions:1) In the standard mode (van Vreeswijk and Sompolinsky, Neural Computation 1998), connectivity is high, and so the diagonal terms, d_i_, are large. In this regime, there is no response reversal. It's an open question exactly how high connectivity is; it certainly isn't infinity, which is what physicists would like it to be, and the effective strength of the connectivity drops as the firing rate drops. However, when firing rates drop, fluctuations become important, and firing rate models become less believable. We're not asking the authors to do full network simulations (although we would suggest that it would be an interesting avenue for future research). However, they should at least comment on this. Even better would be a back of the envelope calculation showing that the connection strengths between populations are in the right range.

The network in our model is not a balanced network in the sense of [van Vreeswijk and Sompolinsky, 98]. In fact, it is dominated by inhibition (i.e. the sum of all the entries of the connectivity matrix is negative).

In the section “Random network model” (Figure 3) we build a network where each population has multiple units and the connections between units are random. In that case the weights are set so that, in average, the input to each unit of population *i* from population *j* is the same as in the population based model. This means that the scaling of the weights in our model is 1/*m* (where *m* is the expected number of connections from population *j* to each unit of population *i*) and not 1/sqrt(*m*) as in [van Vreeswijk and Sompolinsky, 98].

We have added a sentence in the Materials and methods section “Random network model” (first paragraph) to make this point clearer.

2) The authors do a good job citing the relevant literature. However they avoid framing their work in the context of inhibitory stabilized networks (ISNs). ISNs have very strong recurrent excitation that needs to be stabilized by recurrent inhibition (Ozeki et al.), and show the signature of a complex transient before settling in the equilibrium state – reminiscent of the author's Figure 1D. As remarked in the manuscript the sign flip of M_SV_ requires w_EE_ to be sufficiently large. Is the network an ISN? Figure 2 of Litwin-Kumar et al. (2016) extends ISNs to circuits with multiple interneurons subtypes, and they show that if the total inhibition received by E cells reduces under VIP stimulation then the network is an ISN. What regime is the author's model in? Is this a useful label for their network?

For EI networks, the only term of the response matrix that can flip its sign is M_II_ (as analyzed in [Tsodyks et al., 97, Ozeki et al., 09]). In order to have M_II_ < 0 the network has to be an ISN, so in EI networks having response reversal and being ISN are equivalent.

For a network with multiple interneuron types, the equivalence no long holds. The condition to realize M_SV_ > 0 is W_EE_ > d^∞^_E,_therefore the network has to be an ISN. However, the sign of other entries of the response matrix does not depend on whether W_EE_ is larger or smaller than d^∞^_E_, meaning that in general response reversal is not related to ISNs.

In order to clarify this point we have extended the paragraph of the Discussion where we mention ISNs (fifth paragraph). We have also added a short Materials and methods section analyzing the response matrix for EI networks.

3) In Figure 2D, M_EE_ is negative (-0.35), and if we understand things correctly, it's always negative in the high gain regime. Thus, in the high baseline activity state, an input to the pyramidal neuron population will result in a decrease of pyramidal neuron firing rates. This is at odds with most (all?) data sets. The authors remark on this feature in the last paragraph of the subsection “Circuit behavior explained by response matrix”, but do not address the plausibility of this prediction. Do the authors think this result is a problem for their model? More generally, with new parameters can the authors explain the sign flip in M_SV_ without a sign flip in M_EE_ or are these tethered together somehow?

The negative value of M_EE_ in the high baseline scenario is not a feature of the model, but of the particular matrix that we showed. In fact, it is easy to find other matrices for which M_EE_ is always positive.

In order to avoid confusions we have changed the matrix that we used for figures 1 and 2 so that in the current version M_EE_ is always positive.

4) Dipoppa et al. 2016 is important for justifying the model and the authors cite it frequently (and even republish some of its figures). But this paper has not been peer-reviewed (it's a Biorxiv report), giving it the same veridical status as a personal communication or SFN abstract. It's not appropriate for a peer-reviewed manuscript to depend on data that has not been reviewed. In addition, we're not sure how this will affect Dipoppa et al.'s attempts to get their work published. eLife is peer reviewed, and many journals won't let you republish work that's already published in a peer reviewed journal. In this manuscript, the authors actually take figures out of the other group's non-reviewed preprint and publish them in their own paper.It seems to us that Dipoppa et al. is not absolutely essential; Figure 4E could be dropped without affecting the paper much. If the authors do want to include it, they should do two things. First, they should make it crystal clear that Dipoppa et al. is not peer-reviewed, every single time the citation is made. They can leave no doubt in the readers' minds that data is not yet part of the scientific literature. Second, they should get permission from Dipoppa et al. before publishing their data. We're guessing eLife requires this, but even if it doesn't, it's not worth irritating one's colleagues for something that is not essential to one's story.

We would like to thank the reviewers for this important remark. Following their advice, we have removed Figure 4E from the main text and we have included it as a supplement to Figure 4 (Figure 4—figure supplement 1). Furthermore, we have explicitly mentioned that [Dipoppa et al., 16] is a preprint whenever we cited it in the text.

We have also the explicit permission of Dipoppa and his collaborators to present their data in our manuscript.